# Genome-wide analysis of the invasive fungal pathogen *Neopestalotiopsis* reveals high genomic diversity, effector repertoires, and the assessment of fungicide resistance risks

Tika B. Adhikari[1]*, Norman Muzhinji[2]*, Susmita Gaire[1], Prem B. Magar[1], Anju Pandey[1], Swarnalatha Moparthi[1], Frank J. Louws[1]

**1** Department of Entomology and Plant Pathology, North Carolina State University, Raleigh, North Carolina, United States of America, **2** Department of Plant Sciences, University of the Free State, Plant Pathology Division, Bloemfontein, Republic of South Africa

* tbadhika@ncsu.edu (TBA); MuzhinjiN@ufs.ac.za (NM)

## Abstract

*Neopestalotiopsis* species (spp.) have recently emerged as major pathogens world-wide, causing leaf spot, fruit rot, and crown rot of strawberry, resulting in increased disease outbreaks and economic losses. Despite their increasing impact, there is a limited genomic resource for this pathogen group, constraining insights into pathogenic mechanisms and fungicide resistance. We conducted whole-genome sequencing of 50 *Neopestalotiopsis* strains from major strawberry production regions in North Carolina using the Illumina NovaSeq platform. The generated genome assemblies enabled comprehensive analyses of fungicide-resistance-associated mutations and the prediction of effector repertoires using signal-peptide screening, secretion filtering, and machine-learning-based approaches. Eight key fungicide target genes (β-tubulin, *cytb*, *cyp51A*, *cyp51B*, *Bos1,* and SDHI subunits (*sdhB/C/D*) were examined for putative resistance mutations. Phylogenomic analysis based on single orthologs from protein-coding regions and average nucleotide identity (ANI) matrices grouped the strains into two major lineages (Clades A and B) of *Neopestalotiopsis rosae* and a third distinct lineage (Clade C) comprising other *Neopestalotiopsis* spp. Between 879 and 895, CAZymes were predicted per genome, with glycoside hydrolases and auxiliary activity enzymes predominating in plant cell wall degradation. Effector prediction revealed between 87 and 95 putative secreted effectors, including many cysteine-rich proteins and species-specific candidates. Variation in target gene sequence analysis revealed benzimidazole resistance mediated by β-tubulin E198A/K and widespread QoI resistance associated with *cytb* G143A. The *cyp*51 gene exhibited extensive polymorphism, particularly H147Y and N284H, while the L98H variant was rare. No resistance-associated mutations were detected in any SDHI- or Os-1-target genes. This study represents the first comprehensive genomic assessment of *Neopestalotiopsis* strains infecting strawberry in North Carolina,

**Data availability statement:** Data is available from NCBI SubmissionID: SUB15761194 BioProject ID: PRJNA1398581 SRA: SRS27704756-SRS27704805 https://www.ncbi.nlm.nih.gov/bioproject/1398581.

**Funding:** TBA and FJL received funding. North Carolina Department of Agriculture and Consumer Services (NCDA & CS). Grant contract number 25-025-4018 https://www.ncagr.gov/. The funders had no role in study design, data collection and analysis, decision to publish, or preparation of the manuscript.

**Competing interests:** The authors have declared that no competing interests exist.

revealing substantial genomic diversity, various enzymes and effectors involved in host colonization, and mutations linked to resistance to benzimidazole, QoI, and DMI, but not SDHI fungicides. These findings provide a foundation for developing improved, sustainable fungicide-use and disease-management strategies.

---

## 1. Introduction

Strawberry (*Fragaria × ananassa* Duch. ex Roz.) is a valuable specialty crop grown globally, with production exceeding 9.7 million tonnes and contributing over $15.9 billion to the agricultural economy [1]. In the United States, strawberry production was estimated at $3.46 billion in 2024, predominantly from areas in California and the Southeastern U.S. [2,3]. Despite its economic importance, strawberry cultivation is highly susceptible to various biotic stresses, including pathogens such as *Phytophthora* spp., *Botrytis cinerea*, *Colletotrichum* spp., and *Verticillium* spp., among others, which negatively affect yield and fruit quality [4,5]. In recent years, *Neopestalotiopsis* species (spp.), previously considered as weak or opportunistic pathogens, have emerged as a significant threat, causing leaf spot, crown rot, and fruit rot in major strawberry-producing regions [6,7]. A devastating outbreak in Florida in 2017 marked the first major epidemic in the USA linked to aggressive *Neopestalotiopsis* strains, resulting in yield losses of 40–50% and the destruction of over 80 hectares of commercial fields [6]. The newly identified aggressive *Neopestalotiopsis* strains during the 2017 Florida outbreak exhibited greater mycelial growth and spore production in laboratory tests. These strains also caused more severe symptoms on strawberry fruit and leaves than historically observed strains, particularly *N. rosae* [6]. Subsequent outbreaks were reported in Georgia, South Carolina, North Carolina, Virginia, New Jersey, and California [8–12]. In North Carolina, *Neopestalotiopsis* spp. has become an increasingly important pathogen in both nursery and commercial fruit production, primarily due to the movement of infected transplants among production regions [6,13]. Environmental conditions such as high humidity, warm temperatures (25–30 °C), and prolonged leaf wetness further exacerbate disease development conditions that predominate in North America [14,15].

Accurate identification of *Neopestalotiopsis* spp. is crucial for understanding their virulence and host-pathogen interactions. However, the taxonomic complexity of the genus *Neopestalotiopsis*, along with overlapping morphological traits, complicates accurate identification and population monitoring [16,17]. Although molecular diagnostics targeting the β-tubulin and TEF1-α genes have improved species-level identification, precise differentiation at the species level remains inconsistent, particularly when examining strains from diverse geographic regions or host plants [9,16,18]. Furthermore, *Neopestalotiopsis* populations exhibit significant variability in virulence and host specificity, complicating predictions of outbreak severity and the development of effective management strategies [19,20].

The limited availability of genomic resources has hindered a deeper understanding of the pathogenicity mechanisms of *Neopestalotiopsis* strains. To date, only a few

high-quality genome assemblies are available, including chromosome-scale assemblies for highly virulent *Neopestalotiopsis* strains (19–02) and *N. rosae* (13–481) [21], and a draft assembly of *N. rosae* strain ML1664 [22]. These genomes reveal complex sets of secreted proteins, candidate effectors, and gene clusters involved in secondary metabolite biosynthesis, all of which may contribute to host colonization and tissue necrosis [22]. Despite these advancements, effector biology in *Neopestalotiopsis*, as well as the presence, diversity, and functional significance of effector and CAZyme repertoires across strains, remain largely unknown.

Managing *Neopestalotiopsis* with fungicides presents additional challenges. Multisite fungicides such as captan, thiram, and chlorothalonil have demonstrated consistent effectiveness [23]. However, resistance to QoI fungicides, caused by mutations in the *cytb* gene, and resistance to benzimidazoles linked to β-tubulin mutations, has been reported in related *Pestalotiopsis* populations [24,25]. It is hypothesized that similar mutations may be present in *Neopestalotiopsis,* and regional sensitivity suggests that resistance may evolve independently across pathogen populations [26,27]. Therefore, understanding the genomic basis of fungicide resistance, including specific point mutations in genes such as *sdhB, sdhC*, *sdhD*, *cyp51*, *cytb,* and β-tubulin, is crucial for designing sustainable chemical management strategies for this pathogen.

The emergence of *Neopestalotiopsis* spp. as a major pathogen, coupled with the absence of resistant commercial strawberry cultivars and limited genomic and evolutionary information, underscores the urgent need to understand pathogen populations at the molecular level. Whole-genome sequencing (WGS) enables definitive identification, elucidation of effector gene repertoires, assessment of genome plasticity, and allows identification and tracking of mutations associated with fungicide resistance. These genomic insights can enhance IPM strategies for disease management, aid in the development of diagnostic markers, and inform long-term breeding strategies for host resistance.

In this study, we sequenced the whole genomes of 50 *Neopestalotiopsis* strains from symptomatic strawberry plants using the Illumina NovaSeq 6000 platform. We hypothesize that WGS can identify significant genetic variation among strains responsible for recent disease outbreaks in North Carolina and elucidate associations among specific gene variants, pathogen virulence, and the molecular evolution of fungicide resistance in the invasive fungal pathogen *Neopestalotiopsis*. The main objectives of this study were to (i) delineate strain boundaries and population genetic structure, (ii) annotate and classify the effector repertoire, and (iii) identify mutations associated with fungicide resistance in key target genes.

## 2. Methods and methods

### 2.1. DNA extraction and whole genome sequencing

A subset of 50 strains of *Neopestalotiopsis* spp. from various strawberry-producing counties in North Carolina (S1 Table), representative of the temporal, geographic and plant-source data, were selected, and based on morphological characteristics and pathogenicity tests from a previous study [18]. A single-spore pure culture of each strain was cultured on Difco potato dextrose agar (PDA) medium, prepared by mixing 39 g of PDA powder per liter of distilled water (Difco Laboratory, Detroit, MI, U.S.A.), and supplemented with Streptomycin (50 mg/liter of distilled water) (Sigma-Aldrich, MO, U.S.A.). The PDA plates were incubated at 25°C under a 12-hour light/dark photoperiod for two weeks. Three to four mycelia plugs of each strain were cultured in potato dextrose broth (PDB; Sigma-Aldrich, MO, U.S.A.) and incubated for 14 days at 25 °C on an orbital shaker at 150 rpm. Mycelial masses were harvested by vacuum filtration through sterile Miracloth and ground to a fine powder in 2 mL tubes using a tissue homogenizer (Model D1030-E, Beadbug, Benchmark Scientific Inc., (Edison, NJ, U.S.A.) with a 0.64 cm ceramic bead at 400 rpm for 2 minutes. These samples were shipped overnight to SeqCenter (Pittsburgh, PA, U.S.A.).

At SeqCenter, genomic DNA was extracted using the ZymoBIOMICS DNA Kit (Zymo Research Corp., Irvine, CA, U.S.A.) according to SeqCenter's recommended protocols. To obtain high-molecular-weight DNA suitable for whole-genome sequencing, samples were purified using Qiagen Genomic Purification Columns (Qiagen, Germantown, MD, U.S.A.) according to the manufacturer's recommended protocols. DNA quantity and purity were measured using a

NanoDrop spectrophotometer (Thermo Fisher Scientific, U.S.A.), and structural integrity was evaluated using 1% agarose gel electrophoresis. Only high-quality DNA samples, with an $A_{260}/A_{280}$ ratio of approximately 1.8 to 2.0 and no visible degradation, were selected for downstream library preparation and sequencing. Illumina short-read sequencing was performed on all 50 *Neopestalotiopsis* strains. Nextera XT paired-end libraries with approximately 350 bp insert sizes were prepared according to the manufacturer's protocol and sequenced on an Illumina NovaSeq 6000 platform at the SeqCenter [28,29].

## 2.2. Genome assembly and quality assessment

The raw reads were initially evaluated for overall quality using FastQC v 0.12.1 [30]. This assessment included analyses of per-base quality scores, GC content, adapter contamination, and sequence duplication levels. Following this quality assessment, the reads were trimmed with Trimmomatic v 0.39 [31] using the following parameters: *SLIDING-WINDOW:4:20, LEADING:3, TRAILING:3, and MINLEN:50*. Further filtering and quality enhancement were performed with fastp v 0.23.2 [32]. The parameters applied during this step included *–detect_adapter_for_pe, –qualified_quality_phred = 20, –trim_poly_g, and –length_required = 50*, which helped to remove poly-G artifacts, trim low-quality bases, and retain high-quality reads suitable for assembly.

Illumina-only assemblies were generated using SPAdes v3.15 [33], with the careful option enabled to minimize misassemblies. Cleaned reads were then mapped back to the draft assemblies using BWA-MEM [34], and alignments were processed with SAMtools v1.17 [35]. Three iterative rounds of polishing were performed using Pilon v1.24 [36] to correct small insertions and deletions (indels) and base-calling errors. Assembly contiguity and overall quality were assessed using QUAST [37]. Assembly completeness was evaluated with BUSCO v5 using the *sordariomycetes_odb10* lineage dataset [38] to quantify conserved single-copy fungal orthologs.

## 2.3. Phylogenomic analysis

Sequence data were aligned using MAFFT v7 [39] with the L-INS-i algorithm. Alignments were visually inspected and refined in MEGA12 [40], then concatenated for phylogenetic analysis. Maximum-likelihood trees were generated with RAxML v8 [41], employing the GTR + G substitution model and 1,000 bootstrap replicates. Resulting trees were visualized and annotated in iTOL v6 [42]. To complement the multilocus phylogeny and achieve genome-scale resolution, orthologous proteins were identified using OrthoFinder v2 [43]. Single-copy orthologs inferred through the STAG/STRIDE workflow were utilized to reconstruct a robust species tree. Whole-genome relatedness among strains and reference species was quantified using the average nucleotide identity (ANI) metric. ANI values were calculated using the ANIm method (MUMmer-based alignment) and PyANI_plus [44] (derived from pyani [45]), providing independent metrics consistent with widely accepted species-level thresholds. Both single-copy orthologous phylogenomic and ANI values produced congruent results, consistently supporting the species identities of all *Neopestalotiopsis* strains and intraspecific lineages.

## 2.4. Secretome, effector, and CAZymes

Genomic sequences were masked using RepeatMasker [46]. Prior to gene prediction, models were trained with AUGUSTUS [47] using fungal-specific parameters. Resulting proteomes were screened for N-terminal secretion signals with SignalP v6.0 [48], transmembrane helices with deepTMHMM [49], and subcellular localization with TargetP v2.0 [50]. Proteins containing signal peptide, lacking transmembrane domains beyond the signal region, and not targeted to mitochondria or other organelles were classified as putative secreted proteins. Candidate effectors were identified using EffectorP v3.0 [51,52]. Proteins shorter than 300 amino acids and containing four or more cysteine residues were classified as cysteine-rich secreted proteins, which are commonly found among fungal effectors. Comparative analyses of effectors and secretomes across strains were conducted using OrthoFinder v2 [43] to infer orthogroups and assess shared and lineage-specific effector repertoires.

To analyze functional diversity among major phylogenetic groups, six representative strains from three clades were selected based on ANI: two strains from Clade A (*N. rosae* strains NC71 and NC143), one from Clade B (NC100), and three from Clade C (NC62, NC127, and NC140). These strains were used for comparative CAZyme annotation alongside publicly available *Neopestalotiopsis* genomes from the National Center for Biotechnology Information (NCBI). Carbohydrate-active enzymes (CAZymes) were annotated using the dbCAN3 meta server [53], integrating predictions from HMMER [54,55], DIAMOND [56], and HotPeP [57]. A protein was considered a CAZyme if it received a significant hit (e-value < 1e-15, coverage > 0.35) from at least two of the three tools, following the dbCAN3 recommended workflow for high-confidence annotation.

## 2.5. Effector proteins annotation and identification of conserved motifs

Functional annotation of predicted effector proteins was performed using a range of complementary databases and tools. Protein family classification and domain prediction were performed using InterProScan [58], which integrates information from Pfam, SMART, ProSite, TIGRFAMs, and other protein family resources. Orthology-based functional inference and Gene Ontology (GO) annotation were performed using eggNOG-mapper v2 [59]. Similarity of each effector protein to known proteins was evaluated using BLASTp searches against reference protein databases [60], yielding preliminary insights into their functional and evolutionary characteristics. Multiple sequence alignment of predicted effector proteins was conducted using ClustalW [61] to examine evolutionary relationships. A phylogenetic tree was constructed in MEGA12 [40] using the maximum-likelihood method with default parameters to resolve clustering patterns and assess relatedness among effectors. Conserved sequence motifs were identified using the MEME Suite [62], using default parameters to detect statistically significant ungapped motifs across the effector dataset. Effector candidates were further characterized using Gene Ontology (GO) terms and targeted BLASTp searches against the Pathogen–Host Interactions database (PHI-base v4.14) [63]. These searches used an identity threshold of >25% and an E-value cutoff of 1e−5 to identify homologs associated with pathogenicity, virulence modulation, and host-pathogen interactions.

## 2.6. Fungicide resistance mutation profiling

Fungicide resistance-associated genes were screened to identify known and potential mutations linked to major classes of fungicides used in plant disease management (Table 1). Coding sequences for key genes were extracted from predicted protein sequences obtained through each de novo genome assembly via BLAST-based searches [60], implemented using custom Linux command-line workflows: *cytb* (QoI resistance), *tub2* (benzimidazole resistance), *cyp51* (DMIs/azole resistance), succinate dehydrogenase subunits *sdhB*, *sdhC*, and *sdhD* (SDHI resistance), and osmotic sensitivity (Os-1).

**Table 1. List of fungicide group-associated resistant genes and their point mutation assessed in this study.**

| Fungicide class | FRAC[a] group | Associated resistant genes | Point mutation |
|---|---|---|---|
| Quinone outside inhibitor (QoI) | 11 | cytochrome b gene (*cytb*) | G143A |
| Benzimidazole | 1 | β-tubulin gene (*tub2*) | E198A/K, F200Y |
| Succinate dehydrogenase inhibitor (SDHI) | | *SdhB* | NA[b] |
| | 7 | *SdhC* | NA |
| | | *SdhD* | NA |
| Demethylation inhibitor (DMI) | | *Cyp51A* | H147Y, L98H, Q172I, M220V, S297T |
| | 3 | *Cyp51B* | N284H, H178S |
| Os-1 family | E3 | *BOS1* | NA |

[a] Fungicide Resistance Action Committee (FRAC) classification system.

[b] No mutation site detected.

The extracted protein sequences were aligned using MUSCLE v3.8 [64] to facilitate high-confidence comparison of amino acid residues associated with resistance. Positional referencing was standardised by mapping eight key gene alignments to well-characterized species: *CytB* to *Zymoseptoria tritici cytb* (NCBI: AAP81933.1); *Cyp51A* was mapped to *Aspergillus fumigatus* (NCBI: AF338659); *Cyp51B* to *Z. tritici* (NCBI: AY253234); *tub2* to *Aspergillus nidulans* benA (NCBI: M17519); *SdhB, SdhC,* and *SdhD* to *Pyrenophora teres* f. sp. *teres* (NCBI: AMD39384.1, NCBI: AMD39385.1, and AMD39386.1), and *Bos1* to *Botrytis cinerea* (NCBI: AF435964) (Table 1).

Alignments were verified using the Fungicide Resistance Alignment Sequence Tool (FRAST; https://www.frast.com.au/), with species codes sourced from the EPPO database (gd.eppo.int). Each gene was screened for known resistance mutations according to the guidelines of the Fungicide Resistance Action Committee (FRAC) and previous research [65]. The specific substitutions analyzed included G143A in the cytochrome b gene (*cytb*), associated with resistance to QoI fungicides, and E198A/K and F200Y mutations in the β-tubulin gene (*tub2*), linked to benzimidazole resistance (Table 1). Additionally, we examined previously reported alterations in the *SdhB, SdhC*, and *SdhD* genes that are associated with decreased sensitivity to SDH inhibitors (SDHIs). Predicted *cyp51* sequences were also analyzed for common substitutions related to DMI (Demethylation Inhibitor) resistance, including H147Y, N284H, and other amino acid changes (Table 1) that have been implicated in reduced azole binding affinity.

## 3. Results

### 3.1. Strain collection

Strains were obtained from 26 out of 100 counties in North Carolina, representing 17 different strawberry cultivars and lines. Of the strains, 41 were cultured from crowns and nine from symptomatic leaves, on 14 and 8 cultivars, respectively (S1 Table). The strawberry cropping cycle in North Carolina is an annual system with fall planting, winter growth, and early spring harvest (April to early June). The strains were obtained from springs in 2023, 2024, and 2025, and from field plants in the fall of 2024 and 2025. Plant sources were not traceable for a few samples and are therefore not reported here.

### 3.2. Genome assembly statistics

Comparative genome assembly statistics indicate a high level of genomic conservation among the *Neopestalotiopsis* strains analyzed. The genome sizes were generally consistent, ranging from approximately 50.9 Mb (NC127) to 53.8 Mb (NC14) (Fig 1). The GC content was also stable across the dataset, ranging from 48.55% to 52.72%, indicating a moderately balanced nucleotide composition with minimal variation between major phylogenetic clades (Fig 1). All genome assemblies comprised 18 contigs and demonstrated very high contiguity, with N50 values of approximately 6.34 Mb (S2 Table). This suggests that the assemblies are of chromosome-level or near-chromosome-level quality. Additionally, BUSCO completeness scores were consistently high, ranging from 99.1% to 99.4%, confirming that the genomes are highly complete and suitable for further comparative and functional genomic analyses (S2 Table).

### 3.3. Phylogenomic structure of *Neopestalotiopsis* strains

Maximum-likelihood phylogenomic analysis of single orthologs from protein-coding regions of whole genome sequencing identified three well-supported clades (A-C) among *Neopestalotiopsis* strains, each characterized by distinct yet conserved genomic features (Fig 1). Clade A consisted of 28 strains and displayed the greatest diversity and encompassed multiple reference genomes of *N. rosae,* indicating that most examined strains belong to, or are closely related to, the *N. rosae* species complex (Fig 1). Our strain in Clade A clustered together with previously identified highly virulent *N. rosae* 19–02 strains [21]. Clade B comprised 20 strains and formed a distinct lineage separate from Clade A but remained within the broader *N. rosae* complex, which also included previously identified moderately virulent *N. rosae* 13–481 strains (Fig 1) [21]. Clade C was the smallest group and comprised two strains, NC71 and NC127, which clustered closely with reference genomes of *N. cubana*, *N. clavispora*, and *N. formicarum* (Fig 1), suggesting that these strains represent a distinct

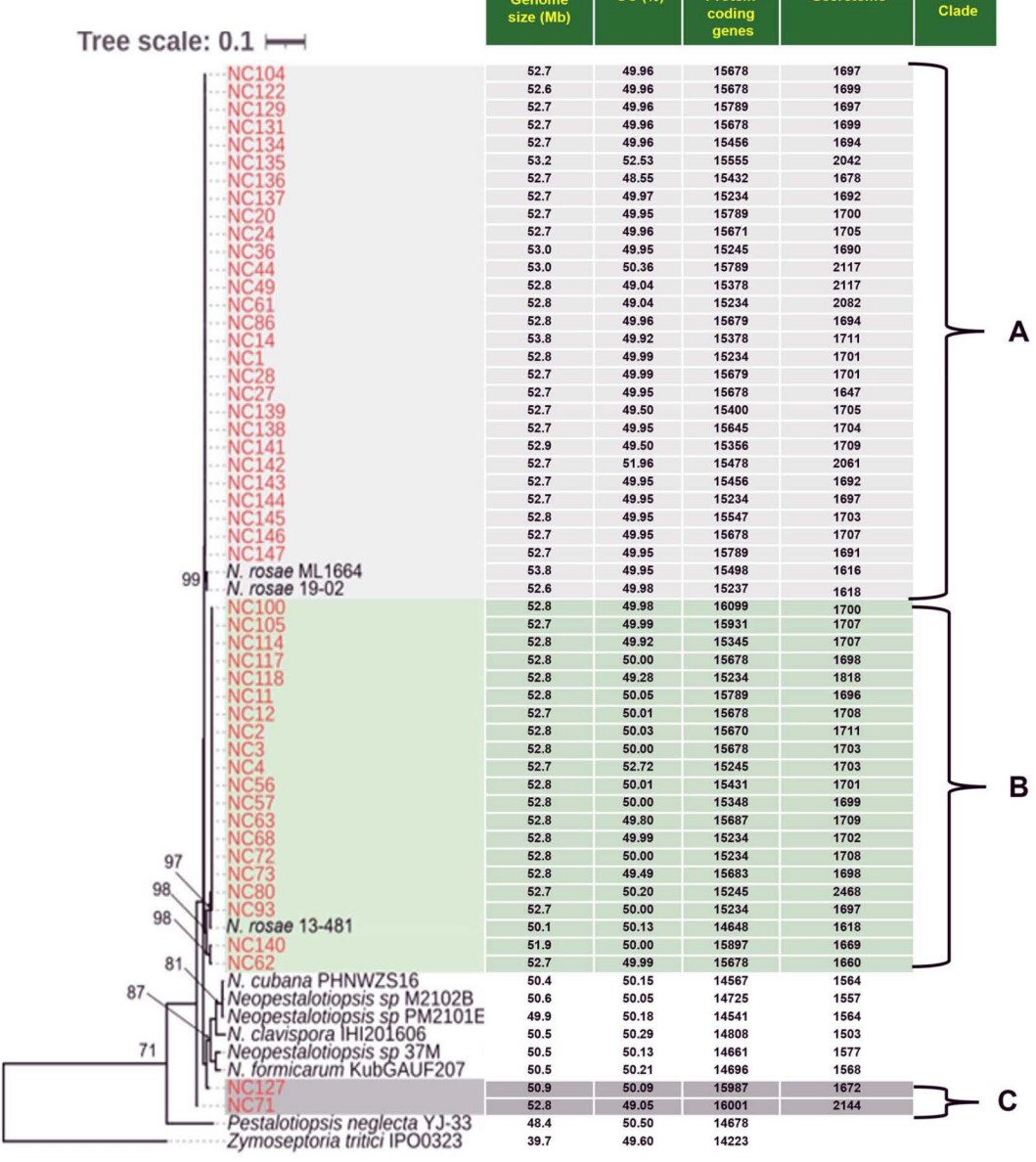

Fig 1. **Phylogenomic relationships and genome features of *Neopestalotiopsis* inferred from single-copy orthologs across whole genomes.** Bootstrap support values greater than 70% are indicated at major nodes. The strains are grouped into three well-supported Clades **(A-C)**, each linked to distinct genomic features, as shown alongside the tree: genome size (Mb), the number of predicted protein-coding genes, GC content (%), and the number of secreted proteins. Clade A comprises most strains and includes several reference genomes of *N. rosae.* Clade B denotes a moderately distinct lineage within the *N. rosae* complex, while Clade C represents a clearly separated group. Outgroup taxa, *Zymoseptoria tritici* and *Pestalotiopsis neglecta*, confirm the monophyly of the main Clades.

species-level lineage within the genus. Strains in Clades A and B exhibited remarkable genomic stability, with the most uniform genome sizes of 52.6–53.8 Mb and 51.9–52.8 Mb, respectively (Fig 1). In contrast, Clade C showed the greatest variation in genome size, ranging from 50.9 to 52.8 Mb, including NC127 (50.9 Mb), the strain with the smallest genome among the 50 examined strains (Fig 1).

The two Clade C strains were isolated from 2 counties: one in the fall of 2024 and the other in May 2025 (the same cropping year), on different cultivars (S1 Table). These strains could not be traced to a plant source to ascertain a possible epidemiological link within the same cropping season. Of particular interest are the three strains from Henderson County, all isolated in the same time period, from 3 different cultivars, with the strains clustering in Clade A (NC122), Clade B (NC118), and Clade C (NC127); 9 strains from Pender county with 2 Clade A strains (NC139 and NC144) and a Clade B strain (NC140) all isolated from the cultivar Brilliance at the same time period; and two strains from Robeson county secured around the same sampling time, representing two different cultivars with one strain in Clade A (NC61) and one in Clade B (NC57) (Fig 1 and S1 Table). These data suggest an epidemic on a given farm may comprise strains from different Clades. Strain groupings were not related to the source of tissue: Clade A comprised seven leaf strains, and 21 crown strains, Clade B one leaf strain and 19 crown strains, and Clade C comprised one of each (Fig 1 and S1 Table).

Average nucleotide identity (ANI)-based comparative genomics provided robust quantitative support for the phylogenomic clades identified via maximum-likelihood analysis using single orthologs in protein-coding regions from whole-genome sequencing (Fig 2). ANI-based clustering was mainly consistent with the maximum-likelihood phylogenetic relationship. However, a few strains (NC62, NC71, and NC140) were clustered differently (Figs 1 and 2). The heatmap displayed clearly defined genomic clusters corresponding to Clades A and B, with consistently high ANI values (>99%) within each cluster, indicating exceptional genomic homogeneity. Both the high nucleotide identity and phylogenetic clustering (Figs 1 and 2, and S3 Table) confirm that these strains comprise two clades that represent a single, closely related lineage, primarily associated with the *N. rosae* species complex. Additionally, the ANI values between strains in Clades A and B consistently ranged from 98% to 99%, demonstrating distinct genomic boundaries between these two clades (Figs 1 and 2, and S3 Table). This separation aligns with the branching pattern identified in the phylogenomic tree, in which Clade B is resolved as a sister lineage to Clade A (Fig 1). Interestingly, NC71 from phylogenetic Clade C showed the highest ANI value (>99%) relative to most strains in Clade A. Furthermore, NC62, NC127, and NC140 had ANI values below 98% with one another and with strains in Clades A and B (Fig 2 and S3 Table). This distinct pattern supports the phylogenomic inference that Clade C represents a separate evolutionary lineage. The outgroup genomes displayed low ANI values (<85%) relative to all *Neopestalotiopsis* strains (Fig 2 and S3 Table), reinforcing their phylogenetic placement outside the focal Clades and validating the monophyly of the examined *Neopestalotiopsis* lineages.

### 3.4. Genome annotation

The distribution of predicted protein-coding genes across strains was consistent, with genomes encoding between 15,234 (NC1, NC61, NC68, NC72, NC93, NC118, NC137, and NC144) and 16,099 (NC100) genes (Fig 1). Clade A genomes exhibited the most stable gene counts. In contrast, strains in Clade C had higher gene counts than most strains in Clades A and B. These findings indicate that the genus *Neopestalotiopsis* maintains a largely conserved genomic architecture, despite being divided into three Clades. Secretome predictions indicated that most strains possess between 1,647 (NC27) and 2,144 (NC71) secreted proteins, with secretome sizes broadly comparable across strains (Fig 1). The NC80 genome in Clade B exhibited an expanded secretome of 2468 genes (Fig 1).

### 3.5. CAZyme repertoire across *Neopestalotiopsis* and related species

The CAZyme repertoires were conserved across the analyzed species; however, *N. rosae* strains from all Clades consistently exhibited the most extensive and diverse array of carbohydrate-active enzymes (Table 1 and S4 Table). The total CAZyme counts in *N. rosae* ranged from 879 to 895, representing a 3–5% increase compared to the other

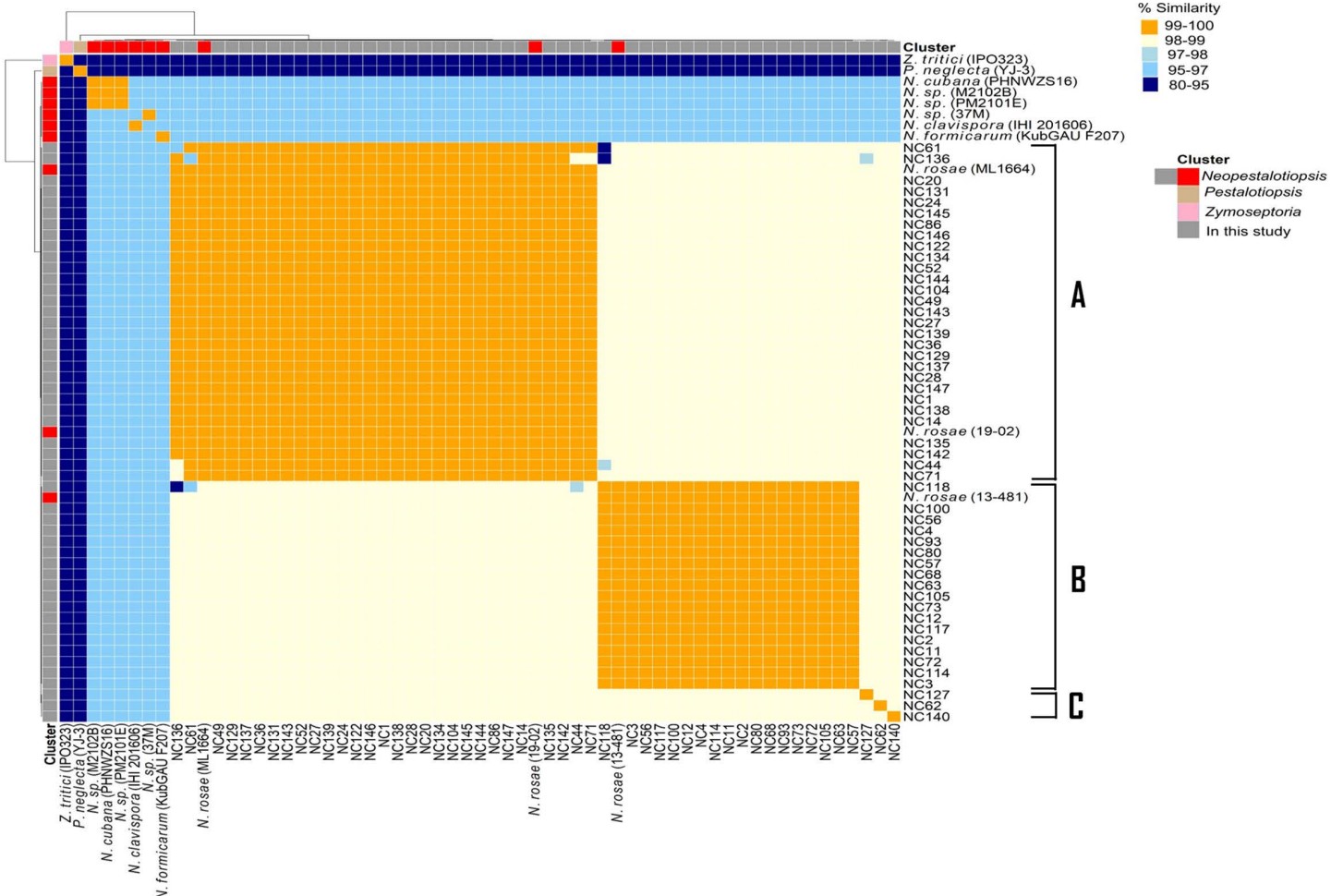

**Fig 2. Heatmap showing pairwise average nucleotide identity among studied strain genomes and reference species.** A hierarchical clustering dendrogram is displayed at the top and left of the table, grouping the genomes by similarity. For percent similarity, yellow indicates high ANI (99-100%), followed by creamy white (98-99%), light blue (95-98%), and dark blue (80-95%). For genome clustering, grey and red indicate *Neopestalotiopsis* strains, with grey explicitly assigned to the genome of the strain from our study and red assigned to the publicly available genome of *Neopestalotiopsis* strains; light brown indicates *Pestalotiopsis*, and pink indicates *Zymoseptoria*.

*Neopestalotiopsis* species (*N. formicarum*, *N. cubana*, and *N. clavispora*) (Table 2 and S4 Table). This enrichment was primarily due to increased numbers of Glycoside Hydrolases (GHs; 401–403) and Auxiliary Activity enzymes (AAs; 237–246) (Table 1 and S4 Table), which are crucial for degrading complex plant cell wall polymers and oxidative breakdown processes. Other CAZyme classes, including Glycosyl Transferases (GTs; 121–125), Polysaccharide Lyases (PLs; 25–29), Carbohydrate Esterases (CEs; 70–75), and Carbohydrate-Binding Modules (CBMs; 20–24) (Table 2 and S4 Table), showed minimal variation among species, indicating a conserved functional backbone across the genus.

In contrast, *N. formicarum*, *N. cubana*, and *N. clavispora* displayed slightly reduced inventories of CAZymes, with total counts ranging from 854 to 860, lower than that of *N. rosae* (Table 2 and S4 Table). These species (*N. formicarum*, *N. cubana*, and *N. clavispora*) showed moderate reductions in GH families (386–387) and AAs (237–240) but maintained proportional representation across other CAZyme classes (Table 2 and S4 Table). Overall, these patterns indicate that while the genus possesses a conserved core of carbohydrate-active enzymes, *N. rosae* contains a more extensive

**Table 2. Distribution of carbohydrate-active enzymes (CAZymes) across different *Neopestalotiopsis* spp.**

| Fungal species | Strain | Glycoside Hydrolases (GHs), | Clade[a] | Glycosyl Transferases (GTs), | Polysaccharide Lyases (PLs) | Carbohydrate Esterases (CEs), | Carbohydrate Binding Modules (CBM), | Auxilary Actvties (AAs) | Total number of CAZymes |
|---|---|---|---|---|---|---|---|---|---|
| *N. rosae* | NC100 | 402 | B | 124 | 28 | 72 | 20 | 246 | 892 |
| *N. rosae* | NC62 | 401 | C | 122 | 27 | 73 | 23 | 237 | 883 |
| *N. rosae* | NC140 | 401 | C | 122 | 28 | 73 | 19 | 242 | 884 |
| *N. rosae* | NC71 | 401 | A | 122 | 27 | 73 | 23 | 237 | 883 |
| *N. rosae* | NC143 | 403 | A | 123 | 25 | 71 | 21 | 242 | 885 |
| *N. rosae* | NC127 | 401 | C | 125 | 29 | 75 | 24 | 241 | 895 |
| *N. rosae* | 1902 | 403 | | 121 | 25 | 70 | 21 | 239 | 879 |
| *N. formicarum* | KubGAU_F207 | 387 | | 116 | 27 | 70 | 17 | 237 | 854 |
| *N. cubana* | PHNWZS16 | 387 | | 116 | 27 | 70 | 20 | 240 | 860 |
| *N. clavispora* | IHI_201606 | 386 | | 121 | 26 | 69 | 16 | 238 | 856 |

[a]ANI-based comparative genomics (Fig 2).

repertoire. This expanded inventory may enhance its ability to colonize host tissues, utilize diverse substrates, and degrade structurally complex plant polysaccharides, traits that could contribute to its ecological versatility and pathogenic potential.

### 3.6. Effector repertoire across *Neopestalotiopsis* and related species

The analysis of secreted effector proteins in *Neopestalotiopsis* strains revealed moderate variability in apoplastic and cytoplasmic effector repertoires (Table 3). In *N. rosae* strains, the total number of effectors ranged from 87 to 95, with apoplastic effectors numbering between 50 and 57, and cytoplasmic effectors ranging from 33 to 39. Strain 19−02 exhibited the highest effector count within *N. rosae*, with a total of 95 effectors (57 apoplastic and 38 cytoplasmic) (Table 3), suggesting a more extensive effector arsenal compared to other *N. rosae* strains. In contrast, *N. formicarum* (KubGAU_F207) and *N. cubana* (PHNWZS16) displayed larger effector complements, with 102 and 103 effectors, respectively (Table 3). Both species showed an increased number of apoplastic and cytoplasmic effectors. On the other hand, *N. clavispora* (IHI_201606) had the lowest count of apoplastic effectors at 48, while its cytoplasmic effector profile was balanced at 43, resulting in a total of 91 effectors (Table 3).

A Neighbor-Joining (NJ) phylogenetic tree was constructed to investigate the evolutionary relationships among the predicted effectors of *Neopestalotiopsis* NC100. This analysis classified 87 effectors into multiple well-supported clusters, highlighting significant sequence diversity within the NC100 secretome (Fig 3). Effectors are generally grouped by function, conserved domain architectures, and motif patterns, suggesting that phylogenetically related proteins may perform similar biological roles. Domain annotation indicated that 14 and 21 effectors contained identifiable conserved domains, as determined by PFAM and InterProScan, respectively (Fig 3 and S5 Table). The remaining effectors were classified as hypothetical or uncharacterized proteins (Fig 3 and S5 Table). Among the annotated effectors, several contained domains closely linked to fungal virulence. For instance, two effectors harbored a cerato-platanin domain, a signature domain associated with loosening plant cell walls, host cell death induction, and fungal pathogenicity. Other effectors possessed domains related to plant cell wall modification, including pectate lyase, glycosyl hydrolase, WSC carbohydrate-binding, peptidoglycan-binding, and RlpA-like domains. This indicates that *Neopestalotiopsis* spp. may utilize a diverse array of enzymes during host infection. Furthermore, motif analysis corroborated the phylogenetic clustering, with distinct motif

**Table 3. Comparative effector repertoires of *Neopestalotiopsis* spp. based on predicted apoplastic and cytoplasmic secreted proteins.**

| *Neopestalotiopsis* spp. | Strain | Clade[a] | Effectors | | |
|---|---|---|---|---|---|
| | | | Apoplastic | Cytoplasmic | Total |
| *N. rosae* | NC140 | C | 56 | 37 | 93 |
| *N. rosae* | NC62 | C | 56 | 36 | 92 |
| *N. rosae* | NC71 | A | 56 | 36 | 92 |
| *N. rosae* | NC143 | A | 53 | 35 | 88 |
| *N. rosae* | NC100 | B | 54 | 33 | 87 |
| *N. rosae* | NC127 | C | 50 | 39 | 89 |
| *N. rosae* | 19−02 | | 57 | 38 | 95 |
| *N. formicarum* | KubGAU_F207 | | 57 | 45 | 102 |
| *N. cubana* | PHNWZS16 | | 60 | 43 | 103 |
| *N. clavispora* | IHI_201606 | | 48 | 43 | 91 |

[a] ANI-based comparative genomics (Fig 2).

compositions recurring within specific branches (Fig 3). Highly conserved motif blocks were observed in groups containing pectate lyases, glycosyl hydrolases, and cerato-platanin proteins, reinforcing the functional coherence of these effector families.

Based on a thorough homology search against the Pathogen-Host Interactions (PHI-base) database, we identified 19 putative virulence-associated genes within the genomic dataset (S6 Table). The analysis revealed significant sequence similarity to experimentally validated pathogenicity factors from various fungal pathogens. For example, NODE_25_length_430131_cov_61.2.g4862.t1 displayed the highest level of conservation, sharing 64.65% amino acid identity with the characterized effector MoCDIP4 from *Magnaporthe oryzae*, which is recognized as a plant avirulence determinant. Other identified homologs included enzymes involved in cell wall degradation, such as cutinases (CUTA; CutA) and pectate lyases (PELD; Pel2), secreted effectors (SRE1; FSE1), and morphogenetic factors (LysM1) (S6 Table). The pathogenic phenotypes associated with the matched entries in the database were diverse: seven hits were linked to reduced virulence (e.g., PELD; MoMAS3), four were effectors (e.g., MoCDIP4; SRE1), and four were associated with increased virulence or hypervirulence (e.g., BAS4; FoSSP1) (S6 Table). Some proteins, such as LysM1, exhibited complex phenotypes related to both increased and reduced virulence (S6 Table). The phylogenetic distribution of the top homologs was broad, indicating close relationships with virulence genes from major pathogenic genera, including *Magnaporthe*, *Fusarium, Botrytis, Verticillium*, and *Lasiodiplodia*. All hits were supported by strong statistical significance, with E-values ranging from 1.09e-90 to 1.20e-08 and bit scores spanning from 29 to 121 (S6 Table).

### 3.7. Fungicide resistance mutation

Screening for resistance-associated mutations across all strains (NC1-NC147) revealed significant diversity in target-site alterations conferring resistance to benzimidazole (MBC), QoI, and DMI fungicides (Table 3). However, no mutations associated with fungicide resistance were detected in any of the SDHI-target genes (SdhB, SdhC, or SdhD) or in the Os-1 family genes (Fig 4A). The β-tubulin mutation E198A/K, linked to high-level MBC resistance, was nearly fixed across the population, present in almost all examined strains (Fig 4A). This suggests that the pathogen is naturally resistant to MBC fungicides. Similarly, the G143A mutation in the *cytB* gene, known to confer strong QoI resistance, was identified in a considerable subset of strains, including NC1, NC11, NC20, NC24, NC27, NC44, NC57, NC62, NC63, NC73, NC80, NC86, NC93, NC104, NC117, NC118, NC122, NC129, NC134, NC136, and NC146. This demonstrates broad

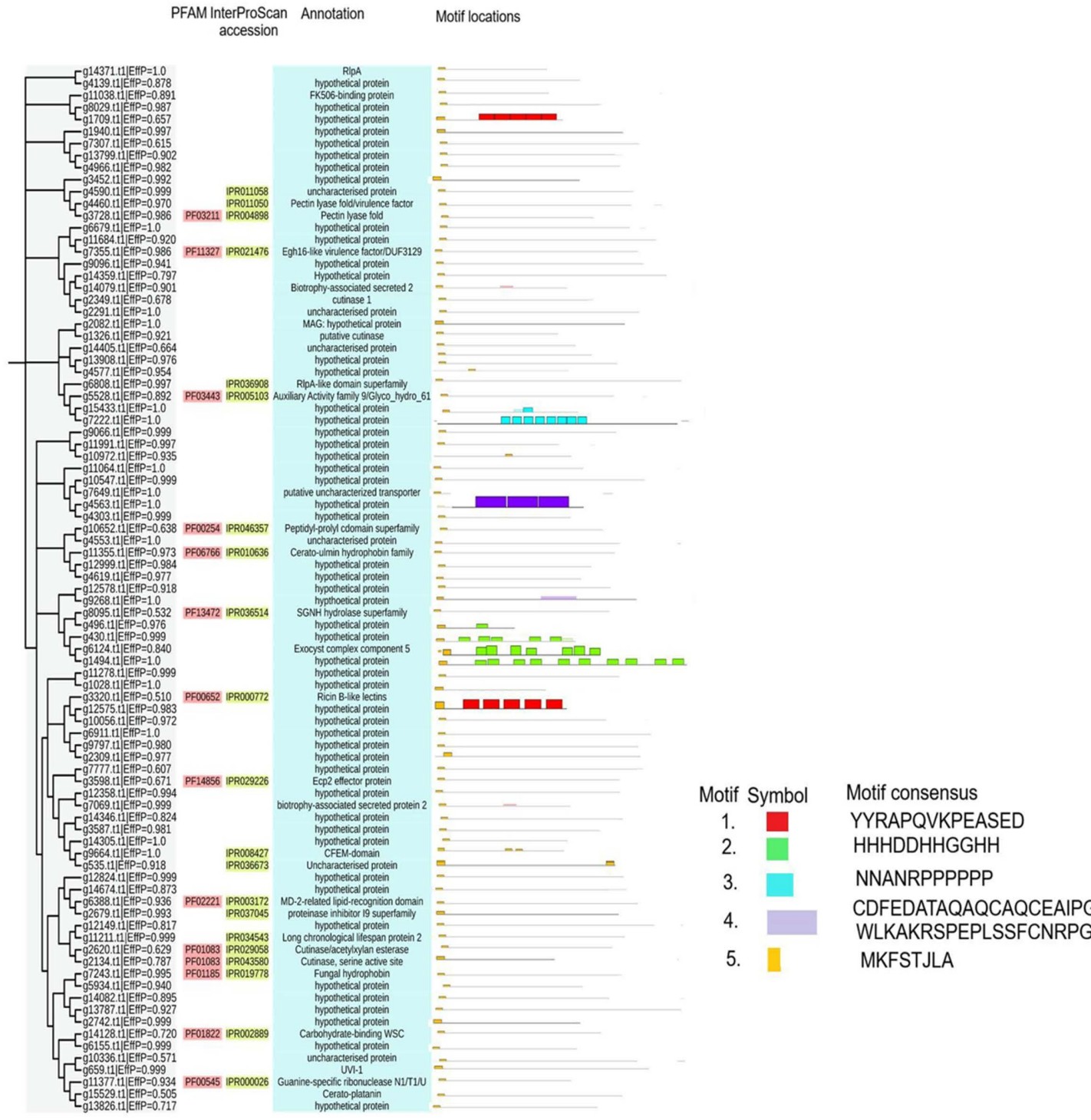

**Fig 3. Characteristics of 87 effector proteins of the *Neopestalotiopsis* NC100 strain.** Various motifs were illustrated using boxes of different colors.

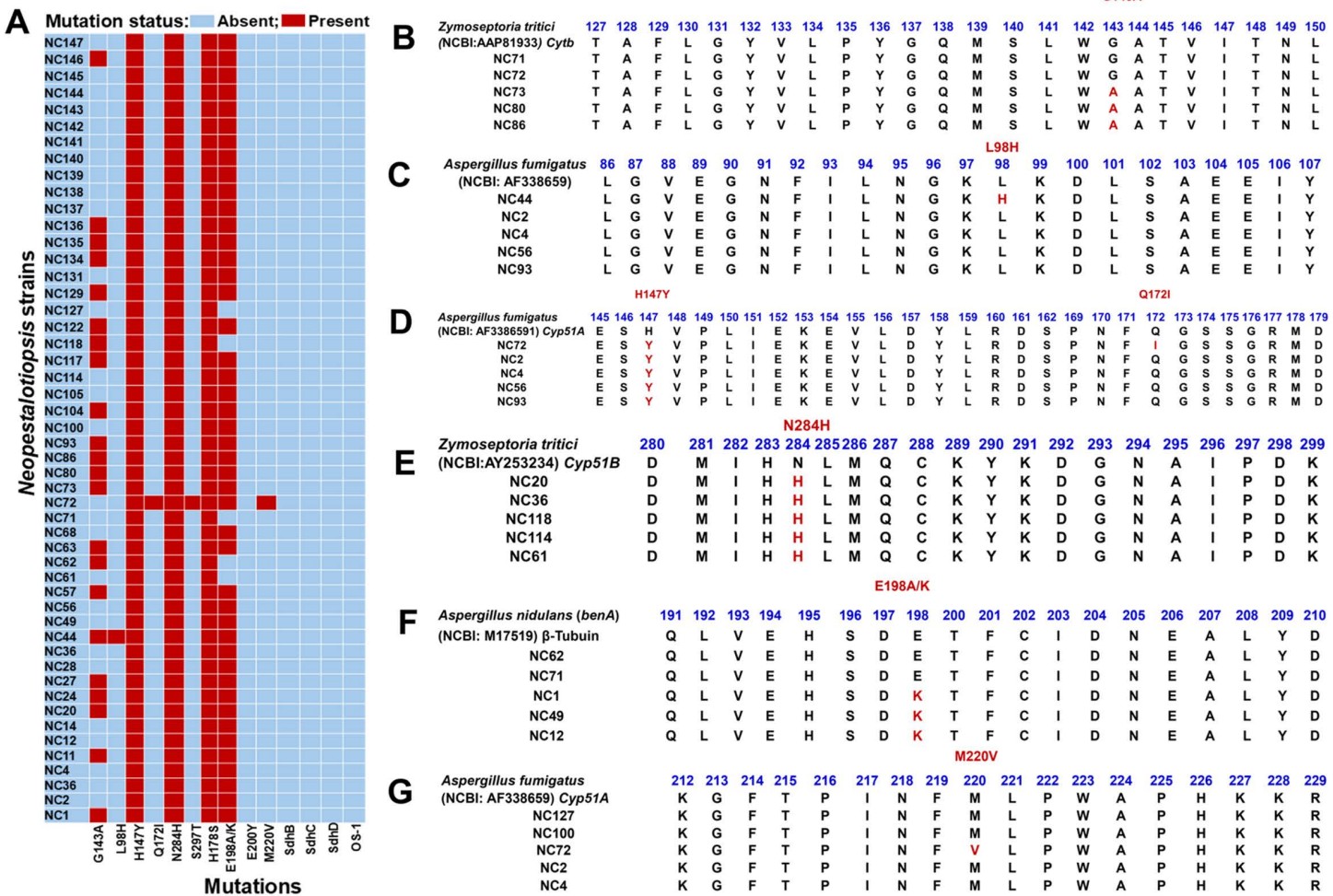

**Fig 4. Summary of fungicide resistance–associated mutations identified across *Neopestalotiopsis* strains.** Panel A: Heatmap summarizes the presence (red) or absence (blue) of known resistance-associated mutations across all 50 sequenced strains (NC1-NC147). The mutations screened include G143A (*cytB*; associated with QoI resistance), L98H, S297T, H147Y, H178S, Q172I, N284H, and M220V (*CYP51A/B*; associated with DMI resistance), E198A/K, and F200Y (*β-tubulin*; associated with MBC resistance), mutations linked to SDHI resistance (*SdhB, SdhC, or SdhD*), and Os_1 family (*BOS1*). It displays amino acid alignments illustrating the specific point mutations identified in individual strains relative to reference sequences. Panel B: *cytB* G143A mutation, Panel C: *CYP51A* L98H mutation, Panel D: *CYP51A* H147Y and Q172I mutation, Panel E: *CYP51B* N284H mutation, Panel F: *β-tubulin* E198A/K mutation, and Panel G: *CYP51A* M220V mutation.

yet incomplete dissemination of the QoI-resistant allele within the population (Figs 4A and 4B). Within the CYP51 family, the DMI-associated mutation H147Y is the most prevalent variant, occurring in all 50 strains (Figs 4A and 4D). The widespread presence of this mutation across such a broad phylogenetic range suggests that H147Y represents a stable, possibly historic, azole-resistant haplotype (Fig 4C). The N284H substitution displayed a similar distribution pattern (Figs 4A and 4E), further supporting the establishment of DMI-resistant alleles. Other CYP51 mutations were infrequent: L98H was observed only in the NC44 strain (Figs 4A and 4C). At the same time, NC72 exhibited a rare multi-mutation DMI-resistant genotype (Q172I, M220V, S297T, and H178S) (Figs 4A, 4D, and 4G). This highlights the ongoing diversification of resistance within specific lineages. Interestingly, no amino-acid substitutions known to confer fungicide resistance were detected in any of the SDHI-target subunits across all strains (Fig 4A), indicating a complete absence of SDHI resistance

markers in this population, despite the widespread distribution of mutations associated with MBC, QoI, and DMI. The presence of highly divergent genotypes, such as the multi-mutant strains NC72 and NC44, demonstrates ongoing adaptive evolution, likely driven by both historical and contemporary fungicide selection pressures.

## 4. Discussion

Traditional morphological tests, along with molecular techniques such as ITS or TEF1-α amplicon gene sequencing, PCR/RFLP, and HRM assays, are valuable for rapid diagnostics [16–19]; however, they have some limitations. Comparative genomics has greatly enhanced our understanding of the genomic regions responsible for the virulence of newly described *Neopestalotiopsis* spp [21,22]. Whole-genome sequencing using advanced platforms such as Illumina and Oxford Nanopore has produced high-quality genome assemblies, which are essential for resolving the complex taxonomy and evolutionary relationships within this fungal pathogen. In this study, Illumina short-read sequencing was used to enable extensive population-level sampling and high sequencing depth across isolates. This approach provides high accuracy and coverage suitable for phylogenomic analyses. The absence of complementary long-read sequencing may limit the resolution of highly repetitive or structurally complex genomic regions. Nevertheless, phylogenomic inference based on single-copy orthologous genes within protein-coding regions offers a robust framework for genome-scale classification and evolutionary analysis. Using this approach, our results demonstrate that the populations responsible for severe outbreaks in NC are genetically distinct, with genome-wide analyses revealing that these isolates cluster into three well-supported lineages or clades.

This study presents the largest genomic resource for *Neopestalotiopsis* strains associated with an emerging strawberry disease. It enables a comprehensive analysis of population genomic diversity, pathogenicity-related gene repertoires, and mutations linked to multiple fungicide resistances. With high-quality genome assemblies (N50 ~6.3 Mb; BUSCO ~99%), our dataset exceeds those of previous studies, which relied on limited genomic data [66], offering valuable insights into pathogen monitoring, fungicide resistance management, and informing breeding strategies for strawberry disease resistance. Phylogenomic and ANI analyses revealed a structured population comprising two dominant lineages (Clades A and B) within *N. rosae* and a third, distinct lineage (Clade C) representing other *Neopestalotiopsis* species. The two major clades A and B represent well-established pathogen lineages that are widely distributed in strawberry-associated environments. This is similar to the dual-lineage structures observed in *Colletotrichum gloeosporioides*, *Botryosphaeria dothidea*, and the *Fusarium oxysporum* species complexes [67–69]. Notably, Clade A comprised highly virulent *N. rosae* 19–02 strains, whereas Clade B included moderately virulent *N. rosae* 13–481 strains [21]. The placement of strains within Clade A therefore suggests a potentially higher level of aggressiveness than those in Clade B. However, phenotypic confirmation is still required. Comprehensive in vitro assays assessing mycelial growth and spore production, together with pathogenicity tests evaluating disease severity of all three Clades (A, B, and C) from our study, are necessary to substantiate its aggressiveness, as previously demonstrated by Baggio et al. [6].

In contrast, the third, minor Clade C comprises genetically distinct strains. Although Clade C is minor, its distinct placement has evolutionary significance. It suggests possibilities such as introgression or hybrid origin, the emergence of a new lineage, or a recent divergence. Hybrid-driven diversification is well documented in species such as *Verticillium*, *Zymoseptoria tritici*, *Magnaporthe oryzae*, and *Fusarium* spp., in which gene flow generates novel virulence types and populations adapted to fungicides [70–72]. Future population-level SNP analysis, mating-type profiling, and recombination assessments will be imperative for ascertaining whether Clade C strains represent incipient speciation or adaptive hybridization, with significant implications for disease epidemics. Clades could not be linked to geographic location, cultivar, or source of plant tissue (leaf or crown tissue). Some evidence suggests that strains from multiple clades may be associated with a given epidemic at a farm site. While the phylogeny suggests distinct lineages, a comprehensive taxonomic revision incorporating morphological characterization, type specimen designation, and multi-locus phylogenetic analysis should therefore be a priority for future investigations to determine whether Clade C constitutes a novel *Neopestalotiopsis* species.

The conserved yet expansive CAZyme repertoire, particularly the enriched glycoside hydrolase (GH) and auxiliary activity (AA) enzymes in *N. rosae*, indicates a strong capacity for plant cell wall degradation, a hallmark of necrotrophic and hemibiotrophic fungi [73]. This enhanced capacity may correlate with more aggressive symptoms in the field. Similar CAZyme-driven virulence has been reported in other pathogens, including *Colletotrichum*, *Botrytis,* and *Sclerotinia* [67,73]. Monitoring CAZyme expression throughout infection stages, particularly at early and late time points, will help elucidate whether this pathogen exhibits hemibiotrophy.

Variations in the presence or absence of several effector families across Clades indicate ongoing diversification, a mechanism recognized as driving host adaptation and virulence evolution in fungal pathogens such as *Magnaporthe oryzae*, *Leptosphaeria maculans*, and *Venturia inaequalis* [71,74,75]. Cerato-platanins are known to induce defense responses and promote host cell death during colonization [76,77]. Meanwhile, NLP toxins act as agents of necrosis, leading to tissue collapse and enabling necrotrophic proliferation [78,79]. Similarly, glycoside hydrolase (GH)-linked secreted effectors contribute to the degradation of host cell walls and the modulation of immune responses, which are strongly associated with necrotrophic phases in pathogens such as *Botrytis cinerea* and *Fusarium oxysporum* [80–82]. In contrast, predicted immune-suppressive small effectors likely assist in early host penetration and biotrophic establishment. The presence of an effector repertoire, which includes cerato-platanins, NLP-like toxins, and GH-associated secreted proteins, in *Neopestalotiopsis* strains further supports a biphasic infection strategy. This strategy involves an initial stealthy biotrophic phase facilitated by putative immunosuppressive effectors, followed by a necrotrophic phase driven by cell wall-degrading enzymes and necrosis-inducing toxins, similar to the life cycle of *Colletotrichum* spp. [67,83,84]. To establish this infection model, functional validation of these candidate effectors will be crucial, including gene knockouts, infiltration assays, overexpression systems, and host transcriptome profiling of key effector candidates. These techniques have proven effective in characterizing effector function in *Magnaporthe oryzae* [84], *Fusarium graminearum* [85], and *Colletotrichum higginsianum* [83].

The presence of numerous homologs to known pathogenicity genes in the PHI-base underscores a conserved core of virulence mechanisms [63]. The widespread occurrence of reduced-virulence homologs across all Clades suggests a conserved core of virulence that is likely essential for infection and disease establishment. Similar patterns of core pathogenicity genes, along with Clade-specific accessory modules, have been observed in other fungi, such as *Fusarium oxysporum* [86], *Magnaporthe oryzae* [71], and *Zymoseptoria tritici* [87]. In these cases, accessory effectors and PHI-validated genes contribute to the diversity of virulence and host adaptation. The identification of homologs associated with hypervirulence is particularly significant, as these genes often increase host susceptibility or accelerate tissue necrosis. Such genes have been central to the aggressiveness observed in the *Botrytis cinerea* and *Colletotrichum* species complexes [68,88]. The abundance of PHI-base virulence factors, along with the presence of CAZyme and effector repertoires, supports a pathogenic strategy that relies on coordinated enzymatic breakdown and manipulation of the host. This approach is characteristic of necrotrophic and hemibiotrophic fungi during disease progression [68,87].

Target-site screening has identified several well-known mutations associated with fungicide resistance, revealing a high-risk resistance profile. One notable example is the *cytb* G143A substitution, which confers high resistance to QoI in populations of *Botrytis cinerea*, *Alternaria alternata*, and *Colletotrichum gloeosporioides* that have been repeatedly exposed to strobilurin applications [89]. We also found variants of β-tubulin E198A/K, which have historically been linked to resistance against benzimidazoles and failures in field control across various fungal pathogens [65,90]. Additionally, several amino acid substitutions in CYP51, such as H147Y and N284H, are frequently associated with azole insensitivity. This insensitivity arises from altered fungicide binding affinity or from overexpression of sterol demethylase genes [91,92]. The presence of these putative mutations across both major Clades indicates that *Neopestalotiopsis* populations have already experienced significant selection pressure due to historical fungicide exposure. This pattern is similar to the resistance evolution observed in *Zymoseptoria tritici*, *Fusarium* species, *Colletotrichum,* and *Botrytis* under intensive spray regimes [65,93,94]. Intriguingly, we did not detect any SDHI-associated and Os-1 family mutations; this absence should

not be interpreted as a guarantee of long-term sensitivity. Resistance to the fungicide group SDHI can develop rapidly following increased usage, often manifesting within 3–5 seasons due to the single-site mode of action and low fitness cost, as seen *in Botrytis cinerea* and *Didymella tanaceti* [95–98].

The identification of substitutions associated with QoI, benzimidazole, and azole resistance in our dataset serves as an early warning that resistance risks are present or being selected for. If not actively managed, this may undermine chemical control strategies. It is important to note that these mutations remain putative molecular markers until they are phenotypically validated, as resistance expression can depend on factors such as gene copy number, efflux pump activity, compensatory pathways, or local environmental conditions [27].

To assess their functional impact, future studies should focus on $EC_{50}$ sensitivity assays, dose-response screening, and expression profiling. Additionally, identifying specific markers will facilitate the development of rapid PCR-based surveillance assays. These assays will be crucial for monitoring allele frequencies across production systems and for guiding effective fungicide rotation strategies.

These findings indicate that *Neopestalotiopsis* is an emerging leaf, fruit, and crown rot pathogen that exhibits early genomic signs of adaptation to fungicides. This underscores the urgent need for proactive management strategies. To slow the development of resistance and maintain the effectiveness of fungicide applications in strawberry disease management programs, it is crucial to integrate genomic surveillance with resistance-risk forecasting, minimize reliance on a single mode of action, and use tolerant strawberry cultivars.

## 5. Conclusion

This study provides the most comprehensive genomic insights into *Neopestalotiopsis*, an invasive pathogen associated with strawberry disease in North Carolina, USA. It reveals that the pathogen system is evolving and is genetically structured into two dominant lineages, along with a third, divergent Clade that may indicate hybridization, recent introductions, or emerging diversification. The convergence of extensive CAZyme repertoires, diverse effector suites, and PHI-validated virulence genes supports the hypothesis of a hemibiotrophic infection strategy that shifts into necrotrophy. This aligns with the observed progression of symptoms in the field and the mechanisms noted in other destructive fruit pathogens. Additionally, the identification of target-site mutations associated with QoI, benzimidazole, and azole fungicides suggests that resistance selection is already underway. Overall, these findings provide a genomic basis for various applications, including molecular surveillance programs, risk forecasting of fungicide resistance, and effector-assisted breeding. As strawberry production increases, integrating genomic monitoring with phenotyping, fungicide management, and host resistance strategies is imperative for sustainable crop production. Our research lays a vital foundation for future studies on the evolutionary pathways of *Neopestalotiopsis* strains, informing sustainable disease management strategies and safeguarding global strawberry production.

## Supporting information

**S1 Table. Fifty strains of *Neopestalotiopsis* species from North Carolina were sequenced in this study.**
(XLSX)

**S2 Table. Genome statistics of *Neopestalotiopsis* species used in this study.**
(XLSX)

**S3 Table. ANI values.**
(XLSX)

**S4 Table. CAZyme repertoire of *Neopestalotiopsis* species.**
(XLSX)

**S5 Table. Effector annotation of NC100 secretome.**
(XLSX)

**S6. Table. PHI database.**
(XLSX)

## Author contributions

**Conceptualization:** Tika B. Adhikari.

**Data curation:** Tika B. Adhikari, Norman Muzhinji, Susmita Gaire.

**Formal analysis:** Norman Muzhinji.

**Funding acquisition:** Tika B. Adhikari, Frank J. Louws.

**Investigation:** Tika B. Adhikari, Susmita Gaire, Prem B. Magar, Anju Pandey.

**Methodology:** Tika B. Adhikari, Norman Muzhinji, Susmita Gaire, Prem B. Magar, Anju Pandey.

**Project administration:** Tika B. Adhikari, Frank J. Louws.

**Resources:** Tika B. Adhikari, Norman Muzhinji, Susmita Gaire, Prem B. Magar, Anju Pandey, Swarnalatha Moparthi.

**Software:** Norman Muzhinji.

**Supervision:** Tika B. Adhikari.

**Visualization:** Tika B. Adhikari, Susmita Gaire.

**Writing – original draft:** Tika B. Adhikari, Norman Muzhinji, Susmita Gaire, Anju Pandey.

**Writing – review & editing:** Tika B. Adhikari, Norman Muzhinji, Susmita Gaire, Prem B. Magar, Anju Pandey, Swarnalatha Moparthi, Frank J. Louws.

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
