## [Decision Letter · Decision Letter 0]

27 Feb 2026

PONE-D-26-00871Genome-wide analysis of the invasive fungal pathogen Neopestalotiopsis unveils high genomic diversity, effector repertoires, and the assessment of fungicide resistance risks.PLOS One

Dear Dr. Adhikari,

Thank you for submitting your manuscript to PLOS ONE. After careful consideration, we feel that it has merit but does not fully meet PLOS ONE’s publication criteria as it currently stands. Therefore, we invite you to submit a revised version of the manuscript that addresses the points raised during the review process.

We look forward to receiving your revised manuscript.

Kind regards,

Pramod Prasad, Ph.D.

Academic Editor

PLOS One

Journal Requirements:

[TBA and FJL received funding.

North Carolina Department of Agriculture and Consumer Services (NCDA & CS).

Grant contract number 25-025-4018

https://www.ncagr.gov/].

5. Thank you for stating the following in your manuscript:

[We express our gratitude to the North Carolina Department of Agriculture and Consumer Services (NCDA & CS) for their financial support through grant contract number 25-025-4018, which facilitated this study.]

[TBA and FJL received funding.

North Carolina Department of Agriculture and Consumer Services (NCDA & CS).

Grant contract number 25-025-4018

https://www.ncagr.gov/]

7. Please remove your figures from within your manuscript file, leaving only the individual TIFF/EPS image files, uploaded separately. These will be automatically included in the reviewers’ PDF.

8. We notice that your figures are uploaded with the file type “Supporting Information”. Please amend the file type to “Figures”.

Additional Editor Comments:

The paper has now been evaluated, and based on the reviewers’ comments and my own assessment, minor revision is recommended. While the study appears scientifically sound and suitable for publication, several minor changes are suggested to improve scientific clarity, coherence, and overall presentation.

Reviewers' comments:

Reviewer's Responses to Questions

**Comments to the Author**

1. Is the manuscript technically sound, and do the data support the conclusions?

Reviewer #1: Partly

Reviewer #2: Yes

2. Has the statistical analysis been performed appropriately and rigorously?

Reviewer #1: Yes

Reviewer #2: Yes

3. Have the authors made all data underlying the findings in their manuscript fully available?

Reviewer #1: Yes

Reviewer #2: Yes

4. Is the manuscript presented in an intelligible fashion and written in standard English?

Reviewer #1: Yes

Reviewer #2: Yes

5. Review Comments to the Author

Reviewer #1: This study presents a timely and comprehensive genomic investigation of Neopestalotiopsis species associated with strawberry diseases in North Carolina, addressing an important gap in genomic resources for this emerging pathogen complex. By sequencing and analyzing 50 strains from major production regions, the authors provide valuable insights into population structure, pathogenicity-related gene content, and fungicide resistance mechanisms.

The paper reads well and could be considered for publication, although addressing several areas for improvement would strengthen it. One such area is the phenotypic validation of fungicide resistance, which would improve the interpretation of resistance-associated mutations, particularly for cyp51 variants linked to DMI resistance. These mutations are more complex than the single, well-established point mutations that govern resistance to benzimidazole and QoI fungicides. Another concern is the reliance on short-read Illumina sequencing without complementary long-read (PacBio or Oxford Nanopore) data, which may limit the resolution of repeat-rich or accessory genomic regions. In addition, the taxonomic status of Clade C warrants clearer resolution, although this may reasonably fall outside the primary scope of the present study. Similarly, the predicted effector repertoire is based solely on in silico analyses and remains hypothetical in the absence of functional or expression-based validation.

Nevertheless, as a pioneering large-scale genomic resource for Neopestalotiopsis, the findings in this paper provide a valuable foundation for future work.

Given the breadth of genome data generated, performing additional analyses such as exploring recombination or horizontal gene flow among lineages could further enhance understanding of population dynamics and the spread of fungicide resistance.

Reviewer #2: Neopestalotiopsis spp. is among the most important emerging strawberry pathogens worldwide. In 2017, this pathogen began to draw attention because of its high impact on crops, as it can attack different plant organs. This paper is a significant contribution that sheds light on various aspects of the pathogen's genomics, explaining why it is aggressive and difficult to control. It provides basic information and links it to an applied solution to manage fungicide resistance.

The paper is very well written. During the review, it called my attention to the fact that the authors no longer use the term "Aggressive and non-aggressive" as used by Baggio et al. (2021. Different groups in the USA have reported the aggressive Neopestalotiops sp., including Canada. Also, recently, Moparthi et al. (2026) reported the identification of different species within the N. rosae complex, including N. hispanica (a controversial report for some groups in the USA). In this paper, they also do not refer to the term "aggressive isolates," as it was called in the last years, for the widespread variant (as it has been called).

I believe the paper should include, in the introduction, a discussion of this information, contrasting how " Aggresive" isolates differed from the rest of N. rosae.

Is clade C, the "aggressive" Neopestalotiopsis? (referred to as N. hispanica? in Moparth et al.) I believe this part needs clarification, given that many reports have emphasized this term.

I strongly suggest that the authors clearly state whether the isolates analyzed fall into the so-called aggressive category or, if they decide that the term is no longer needed based on their results, indicate this in the results and discussion sections.

6. PLOS authors have the option to publish the peer review history of their article (what does this mean?). If published, this will include your full peer review and any attached files.

Reviewer #1: No

Reviewer #2: No

---

## [Author Response · Author response to Decision Letter 1]

24 Mar 2026

March 24, 2026

The Editorial Office, PLOS One

PLOS, 1875 Mission Street

Suite 103 #188 San Francisco, CA 94103

United States

Attn: Dr. Pramod Prasad

Academic Editor

PLOS One

Ref: Submission of revised manuscript ID: PONE-D-26-00871.

We are delighted to have been allowed to revise our manuscript entitled "PONE-D-26-00871 Genome-wide analysis of the invasive fungal pathogen Neopestalotiopsis reveals high genomic diversity, effector repertoires, and the assessment of fungicide resistance risks" (PONE-D-26-00871) for publication in PLOS One. We express our gratitude for the time and effort you all dedicated to providing such valuable guidance. We have diligently considered your feedback, along with the insights from two additional reviewers.

Based on those comments and suggestions, this revision includes several other positive changes.

Clarified portions of the introduction, methodology, and data analysis.

Provided a more exciting yet balanced introduction and discussion based on the study’s results.

Improved and reorganized the paper’s results and discussion for clarity.

Corrected typos and grammatical errors (those several minor comments) to improve the quality of the manuscript.

We hope these revisions improve the manuscript so that you will consider it worthy of publication in PLoS One.

Responses to Journal Requirements

Comment: 4. Thank you for stating the following financial disclosure:

[TBA and FJL received funding].

North Carolina Department of Agriculture and Consumer Services (NCDA & CS).

Grant contract number 25-025-4018

https://www.ncagr.gov/].

Response: TBA and FJL received funding from North Carolina Department of Agriculture and Consumer Services (NCDA & CS). FJL was a principal investigator (PI), and TBA was a co-investigator (Co-PI). This information was already mentioned Author Contributions

Comment: We note that you have provided funding information that is not currently declared in your Funding Statement. However, funding information should not appear in the Acknowledgments section or other areas of your manuscript. We will only publish funding information present in the Funding Statement section of the online submission form.

Please remove any funding-related text from the manuscript and let us know how you would like to update your Funding Statement.

Response: Please enter the following statement in the Funding Statement section of the online submission form.

We express our gratitude to the North Carolina Department of Agriculture and Consumer Services (NCDA & CS) for their financial support through grant contract number 25-025-4018, which facilitated this study.

Responses to Academic Editor's Comments

Comment: The paper has now been evaluated, and based on the reviewers’ comments and my own assessment, a minor revision is recommended. While the study appears scientifically sound and suitable for publication, several minor changes are suggested to improve scientific clarity, coherence, and overall presentation.

Response: We sincerely appreciate your constructive suggestions and valuable insights regarding our manuscript, as well as the opportunity to revise it. After thorough consideration and careful review of the manuscript, along with your feedback and the reviewers', we acknowledge the merit of your recommendations regarding the data analysis and presentation. In response, we have made extensive revisions to the manuscript, enhancing the justification, introduction, relevant figures, data analysis, results, and discussion. We are pleased to inform you that the manuscript has been updated to reflect these improvements. Thank you once again for your guidance.

Responses to Reviewer 1’s Comments

Comment: This study presents a timely and comprehensive genomic investigation of Neopestalotiopsis species associated with strawberry diseases in North Carolina, addressing an important gap in genomic resources for this emerging pathogen complex. By sequencing and analyzing 50 strains from major production regions, the authors provide valuable insights into population structure, pathogenicity-related gene content, and fungicide resistance mechanisms.

Response: We sincerely appreciate the time you dedicated to reviewing our manuscript and the insightful, constructive feedback you provided.

Comment: The paper reads well and could be considered for publication, although addressing several areas for improvement would strengthen it. One such area is the phenotypic validation of fungicide resistance, which would improve the interpretation of resistance-associated mutations, particularly for cyp51 variants linked to DMI resistance. These mutations are more complex than the single, well-established point mutations that govern resistance to benzimidazole and QoI fungicides.

Response: Thank you for bringing up this important point. We agree that phenotypic validation is crucial for fully understanding the role of the potential resistance mutations identified in this study, especially the more complex variants we observed in the cyp51 gene. While our current work focused on establishing a foundational genomic framework for this pathogen population, confirming the functionality of these markers is beyond the scope of this study. We have now highlighted this as a critical next step in the Discussion (Lines 544-549), emphasizing that future research should include in vitro phenotyping assays to validate the relationship between these genetic variants and reduced fungicide sensitivity.

Comment: Another concern is the reliance on short-read Illumina sequencing without complementary long-read (PacBio or Oxford Nanopore) data, which may limit the resolution of repeat-rich or accessory genomic regions. In addition, the taxonomic status of Clade C warrants clearer resolution, although this may reasonably fall outside the primary scope of the present study. Similarly, the predicted effector repertoire is based solely on silico analyses and remains hypothetical in the absence of functional or expression-based validation.

Response: We appreciate the reviewer's insightful comments, which underscore crucial considerations that will shape the future direction of this research. In our study, we acknowledge that while relying on short-read Illumina sequencing has its limitations, using long-read technologies such as PacBio or Nanopore would significantly improve resolution for repetitive and accessory genomic regions. This approach would also facilitate fully closed genome assemblies (Discussion lines 444-456). However, our primary aim was to establish a comprehensive genomic framework for an emerging pathogen population. To achieve this, we sequenced multiple strains (n = 50) to gain insights into population structure and the distribution of key resistance and pathogenicity genes. For these objectives, the depth and cost-effectiveness of Illumina short reads were particularly well-suited, allowing us to confidently capture variation in presence and absence across the core genome. We also agree with the reviewer that the taxonomic status of Clade C merits further study.

Although the reviewer suggests that a complete taxonomic revision may be appropriate, we believe that this is beyond the focus of our population-centric study. Consequently, we have adopted a cautious approach by referring to these groups as "clades" rather than proposing new species designations. This has been clarified in the Discussion (Lines 470-476, 487-490). While the phylogenetic evidence strongly indicates distinct lineages, formal taxonomic descriptions would necessitate additional morphological characterization and ideally, type-specimen designations in accordance with the requirements for species classification.

Lastly, we agree that the predicted effector repertoire is still strictly hypothetical. We have carefully framed these findings as a candidate list for future research rather than confirmed virulence factors. To reinforce this point, we revised the relevant section of the Discussion (Lines 513 and 516) to clarify that these are putative and candidate effectors that need functional validation. This could include techniques such as transient expression assays, gene knockouts, or expression profiling during infection, which are essential next steps to determine which candidates genuinely contribute to pathogenicity.

Comment: Nevertheless, as a pioneering large-scale genomic resource for Neopestalotiopsis, the findings in this paper provide a valuable foundation for future work.

Response: Thank you very much for your thoughtful remarks. We sincerely appreciate your kind words.

Comment: Given the breadth of genome data generated, performing additional analyses such as exploring recombination or horizontal gene flow among lineages could further enhance understanding of population dynamics and the spread of fungicide resistance.

Response: We thank the reviewer for this valuable suggestion. We agree that analyses of recombination events and potential horizontal gene flow among lineages could provide additional insights into the evolutionary dynamics of Neopestalotiopsis populations, including the potential dissemination of fungicide resistance. However, recombination analyses are generally most informative when applied to densely sampled populations across multiple geographic regions, where sufficient genetic variation exists to reliably detect recombination breakpoints and horizontal gene flow. In the present study, the isolates were sampled primarily to resolve lineage-level phylogenomic relationships rather than to capture fine-scale population diversity; therefore, the dataset is not optimally structured for robust inference of recombination dynamics. As a result, such analyses would offer limited additional insight into the primary objectives of this work.

Responses to Reviewer 2’s Comments

Comment: Neopestalotiopsis spp. is among the most important emerging strawberry pathogens worldwide. In 2017, this pathogen began to draw attention because of its high impact on crops, as it can attack different plant organs. This paper is a significant contribution that sheds light on various aspects of the pathogen's genomics, explaining why it is aggressive and difficult to control. It provides basic information and links it to an applied solution to manage fungicide resistance.

Response: We sincerely appreciate the time you took to share your valuable insights and suggestions. Your contributions have played a crucial role in shaping our revision process, and we are truly grateful for your input.

Comment: The paper is very well written. During the review, it called my attention to the fact that the authors no longer use the term "Aggressive and non-aggressive" as used by Baggio et al. (2021. Different groups in the USA have reported the aggressive Neopestalotiops sp., including Canada. Also, recently, Moparthi et al. (2026) reported the identification of different species within the N. rosae complex, including N. hispanica (a controversial report for some groups in the USA). In this paper, they also do not refer to the term "aggressive isolates," as it was called in the last years, for the widespread variant (as it has been called).

Response: Information on aggressiveness and non-aggressiveness was added in the introduction (lines 57-60, 83), results (lines 287-290), and discussion (lines 466-473).

Comment: I believe the paper should include, in the introduction, a discussion of this information, contrasting how " Aggresive" isolates differed from the rest of N. rosae. Is clade C, the "aggressive" Neopestalotiopsis? (referred to as N. hispanica? in Moparth et al.) I believe this part needs clarification, given that many reports have emphasized this term. I strongly suggest that the authors clearly state whether the isolates analyzed fall into the so-called aggressive category or, if they decide that the term is no longer needed based on their results, indicate this in the results and discussion sections.

Response. Information regarding aggressiveness and non-aggressiveness has been incorporated into the introduction (lines 57-60), the results (lines 287-290), and the discussion (lines 466-473).

Again, we wish to express our sincere gratitude for the valuable feedback you have provided. Each comment has been thoughtfully evaluated, and we have incorporated the necessary revisions in the updated manuscript.

Thank you for your attention, and we eagerly anticipate your favorable response.

With best regards,

Tika Adhikari, Ph. D.

Corresponding author

Norman Muzhinji, Ph. D.

Corresponding author

---

## [Decision Letter · Decision Letter 1]

21 Apr 2026

Genome-wide analysis of the invasive fungal pathogen Neopestalotiopsis unveils high genomic diversity, effector repertoires, and the assessment of fungicide resistance risks

PONE-D-26-00871R1

Dear Dr. Adhikari,

We’re pleased to inform you that your manuscript has been judged scientifically suitable for publication and will be formally accepted for publication once it meets all outstanding technical requirements.

Kind regards,

Pramod Prasad, Ph.D.

Academic Editor

PLOS One

Additional Editor Comments (optional):

The major concerns are effectively addressed in the revised version. The manuscript is now clear, rigorous, and impactful. The article can be accepted for publication.

Reviewers' comments:

Reviewer's Responses to Questions

**Comments to the Author**

1. If the authors have adequately addressed your comments raised in a previous round of review and you feel that this manuscript is now acceptable for publication, you may indicate that here to bypass the “Comments to the Author” section, enter your conflict of interest statement in the “Confidential to Editor” section, and submit your "Accept" recommendation.

Reviewer #2: All comments have been addressed

2. Is the manuscript technically sound, and do the data support the conclusions?

Reviewer #2: Yes

3. Has the statistical analysis been performed appropriately and rigorously?

Reviewer #2: Yes

4. Have the authors made all data underlying the findings in their manuscript fully available?

Reviewer #2: Yes

5. Is the manuscript presented in an intelligible fashion and written in standard English?

Reviewer #2: Yes

6. Review Comments to the Author

Reviewer #2: The findings reported in this paper will contribute to a better understanding of why N. rosae is so aggressive toward strawberries, and why is it so difficult to manage? The point mutations reported herein also explain and complement the findings of other groups currently working on this pathogen.

No more comments. Just correct a minor typo in the abstract: " Neopestalotiopisrosae". Row 14

7. PLOS authors have the option to publish the peer review history of their article (what does this mean?). If published, this will include your full peer review and any attached files.

Reviewer #2: No

---

## [Editor Report · Acceptance letter]

PONE-D-26-00871R1

PLOS One

Dear Dr. Adhikari,

I'm pleased to inform you that your manuscript has been deemed suitable for publication in PLOS One. Congratulations! Your manuscript is now being handed over to our production team.

Kind regards,

on behalf of

Dr. Pramod Prasad

Academic Editor

PLOS One